

# The propagator matrix reloaded

**João F. Melo**

DAMTP, Centre for Mathematical Sciences,
University of Cambridge, Wilberforce Road, Cambridge CB3 0WA, UK

jfm54@cam.ac.uk

## Abstract

The standard way to perform calculations for quantum field theories involves the S-matrix and the assumption that the theory is free at past and future infinity. However, this assumption may not hold for field theories in non-trivial backgrounds such as curved spacetimes or finite temperature. In fact, even in the simple case of finite temperature Minkowski spacetime, there are a lot of misconceptions and confusion in the literature surrounding how to correctly take interactions into account when setting up the initial conditions. The objective of this work is to clear up these misconceptions and provide a clean and simple derivation of a formalism which includes interactions in the initial conditions and assesses whether or not it is legitimate to ignore them. The ultimate conclusion is that we cannot ignore them: quantum field theories at finite temperature are not free in the infinite past.

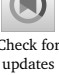
# 1  Introduction

The S-matrix is the usual object of interest when performing calculations in quantum field theory [1–4]. It has been extremely successful at reproducing experimental results in particle accelerators but it presents a challenge: in order to construct the 'in' and 'out' aymptotic states we need to assume the theory is asymptotically free at future and past infinity. This is perfectly justified for zero temperature Minkowski spacetime: if we consider local interactions and the 'in' and 'out' states are spatially well separated we do expect the interactions to die off. However, this might not be the case if we are in the presence of a background field, such as curved spacetime, or are studying a thermal state. In this case the thermal bath and/or the background will keep interacting with the particles possibly ruining our physical picture.

In order to get around these issues we need to calculate new observables, ones which allow us to probe these regimes without assuming the theory is free and checking whether or not our assumptions work. This is precisely what is accomplished by the Schwinger-Keldysh formalism (also sometimes called the 'in-in' formalism) [5,6]. In this formalism we return to the picture most common in undergraduate quantum mechanics: setting up an initial state at time $t_0$, evolving up to time $t$, and calculating the expectation value of the relevant operator. As long as we have control over the initial state, and have the technical prowess to perform the time evolution and evaluate the expectation, there is no need to assume the interactions decay at any time.

This formalism has become a standard tool, being the topic of several textbooks and reviews [7–20]. Using this tool, a lot of attention has been devoted towards studying the situation in the far future. In this case, the main phenomenology is that of secular growth, that is, loop corrections which grow linearly in time, seemlingly ruining perturbation theory at late times. These kinds of issues are well known in the finite temperature literature [9–12, 18, 21–28] and in the case of de Sitter spacetime [29–42]. There have been some calculations performed in black hole and Rindler scenarios [26, 42–47] but the status is less clear in these cases. In order to handle these divergences one needs to construct a modified effective field theory which can take into account the open system character of theories at finite temperature and in the presence of event horizons [29,30,33,34,37–40]. Studying these divergences was the original motivation for this paper and will be the subject of an upcoming publication [48].

However, considerable less attention has been devoted to what happens in the far past. In fact it seems like there is a lot of misunderstanding and confusion in the literature regarding how to appropriately set up initial conditions. Many of the common textbooks and reviews just assume the theory is free at past infinity, essentially ignoring the issue [7,8,10,11,13–16, 18,19,21,22,26]. Some works are more detailed but end up either changing the dynamics explicitly to turn off the interactions [17,41,49,50] or are based in [51,52] (for example, [9,12,23,24,53,54]) whose arguments have a number flaws which will be discussed in the main body of the paper and in the conclusion.

The objective of this paper is to clear up these misconceptions and provide a clean and simple derivation of a formalism which includes interactions in the initial conditions and assesses whether or not it is legitimate to ignore them. The ultimate conclusion is that we cannot ignore them. There are a number of issues with the standard treatments and explicitly computing the 4-point function one can see that it is never turned off. Quantum field theories at finite temperature are not free in the infinite past.

The manuscript is structured as follows:

In section 2, we begin with a brief overview of the Schwinger-Keldysh path integral at a level which should be accessible to readers not familiar with this formalism. We shall pay close attention to the non-triviality of the temporal boundary conditions and the appearance of additional field variables, both of which characteristic of this technique.

In section 3, we detail the construction of the Feynman rules for finite temperature initial conditions. We are vary careful about our assumptions and detailed in our reasoning, in particular we shall not assume the interactions are turned off at past infinity and shall set initial conditions at a finite time in the past $t_0$. The natural conclusion of this calculation is the appearance of a $3 \times 3$ propagator matrix, including mixing between the real-time and imaginary-time field variables.

In section 4, we continue our analysis by computing the symmetric propagator up to one-loop in an on-shell subtraction scheme. This is correlation function which is necessary to determine the energy-momentum tensor and therefore it has clear physical significance. We pay close attention to the role of the cross terms in our calculation and how the most common approaches in the literature would fail or succeed in obtaining the correct answer.

The conclusion is that the $3 \times 3$ approach is more mathematically well-defined and much more straightforward at obtaining the physical answer. However, when resumming the poor IR behaviour of this correlator we find an agreement with the standard approaches. An interpretation for this is provided, nevertheless, this means this calculation is not entirely conclusive on its own regarding the fate of interactions in the far past.

In section 5, we settle the question by computing the equal-time 4-point function at tree-level for a particular choice of external momenta. The result is unambiguous: the $3 \times 3$ propagator matrix is essential to reproduce the correct answer. Not only is the outcome completely independent of time (which on its own implies the interactions are finite at all times) but also the final answer comes purely from the cross terms.

In section 6, we conclude by contrasting with the different approaches found in the literature.

There are also three appendices. The first details the subtleties of the temporal boundary conditions for the time derivative of our fields. The second discusses minus signs and factors of i for Feynman rules mixing real and imaginary time. The third and final one includes some extra calculations for the other propagators which confirm the picture suggested by the symmetric propagator.

**Notation and conventions:** We use the 'mostly-plus' sign convention for the Minkowski metric $\eta_{\mu\nu} = \text{diag}(-++\dots)$. Greek indices $\mu, \nu, \dots$ go from 0 to $d$ and Latin indices $i, j, k, \dots$ go from 1 to $d$, where $d$ is the number of spacial dimensions (usually 3), so that $D = d + 1$ is the number of spacetime dimensions (usually 4). We also use boldface $\boldsymbol{p}$ for purely spatial vectors.

## 2 Review of the Schwinger-Keldysh path integral

In its essence the Schwinger-Keldysh formalism [5–11, 13–22, 41, 49, 53, 55, 56] (also known as 'in-in' formalism) is an initial value formulation of quantum field theory. Instead of considering an 'in' state, $|\text{in}\rangle$, at past infinity and an 'out' state, $|\text{out}\rangle$, at future infinity to then compute the transition amplitude, $S = \langle\text{out}|\text{in}\rangle$; we set up an initial state, $|\psi(t_0)\rangle$, time evolve it, $U(t_f, t_0)|\psi(t_0)\rangle$, and then compute the expectation value of some operator $\mathcal{O}(t_f)$:[1]

$$\langle\mathcal{O}(t_f)\rangle_\psi = \langle\psi(t_0)|U^\dagger(t_f, t_0)\mathcal{O}(t_f)U(t_f, t_0)|\psi(t_0)\rangle. \tag{2.1}$$

The only difference between this formalism and the usual one is what we are calculating. We can apply this formalism for any theory and any initial state if what we are interested in are expectation values of operators at some time $t_f$. However, it is worth noting that this formalism is especially useful for time-dependent or out of equilibrium calculations.

In order to perform concrete calculations we need to convert Eq. (2.1) to a path integral. To accomplish this we begin by inserting the identity many times:[2]

$$s\langle\psi(t_0)|U^\dagger(t_f, t_0)\mathcal{O}(t_f)U(t_f, t_0)|\psi(t_0)\rangle =$$
$$\int\left(\prod dq_i\right)\langle\psi(t_0)|q_1\rangle\langle q_1|U^\dagger(t_f, t_0)|q_2\rangle\langle q_2|\mathcal{O}(t_f)|q_3\rangle\langle q_3|U(t_f, t_0)|q_4\rangle\langle q_4|\psi(t_0)\rangle. \tag{2.2}$$

Let us analyse each factor in turn:

- $\langle q_3|U(t_f, t_0)|q_4\rangle$ is an ordinary path integral with a finite time interval and fixed temporal boundary conditions. The derivation of this fact can be found in standard textbooks and reviews [1–4, 14, 20, 57].

- $\langle q_1|U^\dagger(t_f, t_0)|q_2\rangle$ is also an ordinary path integral, however, the presence of the $U^\dagger$ means we are evolving backwards in time from $q_2$ at $t_f$ to $q_1$ at $t_0$. This means we will get an integrand of $e^{-iS}$ instead of the more familiar $e^{iS}$.

- If our operator of interest is a product of fields (as we shall assume for the remainder of this manuscript) then $\langle q_2|\mathcal{O}(t_f)|q_3\rangle \propto \delta(q_2 - q_3)$, and therefore $q_2 = q_3$ and the boundary conditions from our two path integrals match at $t_f$.

- Finally, $\langle\psi(t_0)|q_1\rangle$ and $\langle q_4|\psi(t_0)\rangle$ are the initial and final wavefunctions. They cannot be readily converted to a path integral. We need to be careful and integrate over all possible boundary conditions at $t_0$ weighted by these wavefunctions before proceeding. We need to know the functional form of our initial state to perform these calculations.

Putting it all together we get the following path integral:

$$\langle\mathcal{O}(t_f)\rangle_\rho = \int dq_-^0 \, dq_+^0 \, \rho(q_+^0, q_-^0)\int\mathcal{D}q_+\mathcal{D}q_-\mathcal{O}(t_f)e^{iS[q_+]-iS[q_-]}, \tag{2.3}$$

with $q_+(t_0) = q_+^0$, $q_-(t_0) = q_-^0$, $q_+(t_f) = q_-(t_f)$ and where we have generalised to an arbitrary density matrix $\rho$ as the above reasoning carries through with no subtleties.

In essence we are starting at time $t_0$, evolving up to time $t_f$, inserting the operator of interesting, then evolving backwards towards $t_0$, integrating over all possible boundary conditions at $t_0$ weighted by the initial wavefunction. This is sometimes called the 'closed' time contour, however, we should note that it isn't really closed as the fields aren't matched at $t_0$.

---

[1] We are using the Schrödinger picture, the argument in the operator is an explicit time dependence, not a dynamic/Heisenberg time dependence.

[2] We will sometimes use quantum mechanical notation for simplicity, it should be straightforward to extend to quantum field theories.

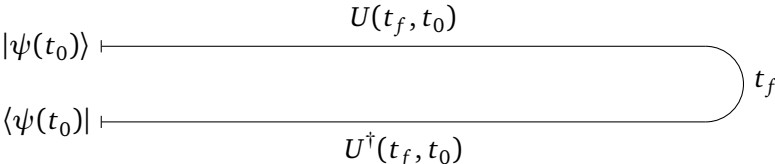

Figure 1: 'Closed' time contour.

A few remarks are in order. Firstly, that we could insert a $U^\dagger(t_{f_2}, t_f)U(t_{f_2}, t_f)$ to get either:

$$\langle\psi(t_0)|\,U^\dagger(t_{f_2}, t_0)U(t_{f_2}, t_f)\mathcal{O}(t_f)U(t_f, t_0)|\psi(t_0)\rangle\,, \tag{2.4}$$

or

$$\langle\psi(t_0)|\,U^\dagger(t_f, t_0)\mathcal{O}(t_f)U^\dagger(t_{f_2}, t_f)U(t_{f_2}, t_0)|\psi(t_0)\rangle\,. \tag{2.5}$$

Therefore, we can actually insert our operator anywhere on the contour. The time where we turn around and match between the forwards and backwards moving fields is merely a bookkeeping parameter and should drop out of the final answer. The physical time variables are $t_0$ when we set our initial conditions and $t_f$ when we insert the operator.

Secondly, we get a doubling of our field variables. Nevertheless, given the actions are just added together, there seems to be no quadratic mixing and we would naively expect two independent propagators. However, the matching $q_+(t_f) = q_-(t_f)$ actually induces a mixing between the two variables and we get a non-diagonal $2 \times 2$ matrix of propagators.

Finally, given we have to integrate over all possible boundary conditions at $t_0$ we cannot integrate by parts to complete the square as is usual, we have to be a bit more careful. A particularly pedagogical overview of how to perform this for a free theory (including finite temperature and excited states) can be found in [20].

## 3 Tree-level propagators

In this section, we will describe how to construct the Schwinger-Keldysh style path integral, using a finite temperature initial density matrix, set at a finite time in the past, and without assuming the theory to be free at any time. We end by presenting the corresponding Feynman rules for a $\phi^4$ theory.

### 3.1 The finite temperature path integral

The finite temperature density matrix is a particularly simple state to construct at any time and without assuming the theory to be free. This is because it is straightforward to convert it to a path integral. We just have to note that the usual Gibbs state (where $\beta$ is the inverse temperature, $H$ is the Hamiltonian and we have ignored the normalisation as its only role is to cancel the vacuum bubbles):

$$\rho = e^{-\beta H}\,, \tag{3.1}$$

can be written as a time evolution, albeit in an imaginary direction,

$$\rho = e^{-\beta H} = U(t_0, t_0 - i\beta)\,, \tag{3.2}$$

where, for a time dependent Hamiltonian, we should evaluate it at time $t_0$. This can be readily converted to a Euclidean path integral.

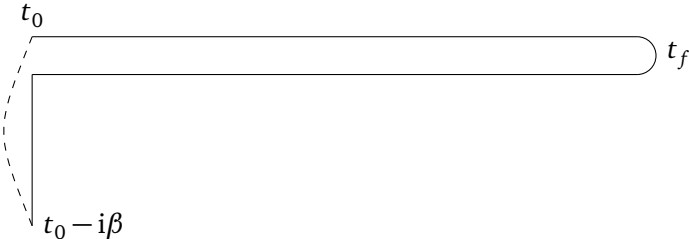

Figure 2: Finite temperature time contour.

The integration over $q_1$ and $q_2$ in Eq. (2.2) then implies that the field values are matched along a contour that includes a segment in an imaginary direction as is shown in Fig. 2.

Our path integral then looks like (for the quantum mechanical theory):

$$Z = \int \mathcal{D}q_+ \mathcal{D}q_- \mathcal{D}q_E \, e^{iS[q_+]-iS[q_-]-S_E[q_E]}, \tag{3.3}$$

where

$$S[q_\pm] = \int_{t_0}^{t_f} dt \left( \frac{1}{2}\dot{q}_\pm^2 - \frac{1}{2}m^2 q_\pm^2 + J_\pm q_\pm \right), \tag{3.4}$$

$$S_E[q_E] = \int_0^\beta d\tau \left( \frac{1}{2}q_E'^2 + \frac{1}{2}m^2 q_E^2 + J_E q_E \right), \tag{3.5}$$

and where $\tau = -it$ is a real parameter for the imaginary segment, $\dot{}$ represents derivatives with respect to $t$ and $'$ derivatives with respect to $\tau$. We have also included sources in anticipation of the calculations to follow and to be more explicit about the sign convention for the factors in front of the sources.

As is clear from the canonical construction for the Schwinger-Keldysh path integral we should impose the following boundary conditions:

$$q_+(t_f) = q_-(t_f), \tag{3.6a}$$

$$q_-(t_0) = q_E(0), \tag{3.6b}$$

$$q_+(t_0) = q_E(\beta). \tag{3.6c}$$

Slightly less obviously we should also impose boundary conditions on the time derivatives of the fields. This will be necessary to solve the propagator equations as they will involve second time derivatives. As is argued in App. A we are free to choose these to be whatever we want. For simplicity we then choose the time derivatives such that all the boundary terms cancel when we integrate by parts:

$$\dot{q}_+(t_f) = \dot{q}_-(t_f), \tag{3.7a}$$

$$\dot{q}_-(t_0) = iq_E'(0), \tag{3.7b}$$

$$\dot{q}_+(t_0) = iq_E'(\beta). \tag{3.7c}$$

Note that $i\frac{d}{d\tau} = \frac{d}{dt}$ which gives some intuition for the factor of i in these equations.

## 3.2 The propagator equations

In order to derive the Feynman rules for $\phi^4$ theory in $D$-dimensional Minkowski spacetime we need to first compute the quadratic path integral including sources. By Fourier transforming

in the spatial directions we get the same as in Eq. (3.4) where the coefficient in front of the quadratic term is replaced by $E_p = \sqrt{p^2 + m^2}$ where $p$ is the spatial momentum, and $m$ is the mass of the particle. Due to this we will continue to use quantum mechanical notation, knowing that it is equivalent to $D$-dimensional Minkowski spacetime.

Our path integral then looks like, after integrating by parts,

$$
\begin{aligned}
Z = \int \mathcal{D}q_+ \mathcal{D}q_- \mathcal{D}q_E \exp\Bigg\{ & i \int_{t_0}^{t_f} dt \left[ -\frac{1}{2} q_+(t) \left( \frac{d^2}{dt^2} + m^2 \right) q_+(t) + J_+(t) q_+(t) \right] \\
& - i \int_{t_0}^{t_f} dt \left[ -\frac{1}{2} q_-(t) \left( \frac{d^2}{dt^2} + m^2 \right) q_-(t) + J_-(t) q_-(t) \right] \\
& - \int_{0}^{\beta} d\tau \left[ -\frac{1}{2} q_E(\tau) \left( \frac{d^2}{d\tau^2} - m^2 \right) q_E(\tau) + J_E(\tau) q_E(\tau) \right] \\
& + i \left[ \frac{1}{2} q_+(t) \dot{q}_+(t) \right]_{t_0}^{t_f} - i \left[ \frac{1}{2} q_-(t) \dot{q}_-(t) \right]_{t_0}^{t_f} \\
& - \left[ \frac{1}{2} q_E(\tau) q_E'(\tau) \right]_{0}^{\beta} \Bigg\}.
\end{aligned}
\tag{3.8}
$$

Now we need to complete the square. We do the following change of variables:

$$
\begin{aligned}
Q_+(t_1) = q_+(t_1) & + \int_{t_0}^{t_f} dt_2 \, G_{++}(t_1, t_2) J_+(t_2) - \int_{t_0}^{t_f} dt_2^\star \, G_{+-}(t_1, t_2^\star) J_-(t_2^\star) \\
& + i \int_{t_0}^{t_f} d\tau_2 \, G_{+E}(t_1, \tau_2) J_E(\tau_2),
\end{aligned}
\tag{3.9a}
$$

$$
\begin{aligned}
Q_-(t_1^\star) = q_-(t_1^\star) & + \int_{t_0}^{t_f} dt_2 \, G_{-+}(t_1^\star, t_2) J_+(t_2) - \int_{t_0}^{t_f} dt_2^\star \, G_{--}(t_1^\star, t_2^\star) J_-(t_2^\star) \\
& + i \int_{t_0}^{t_f} d\tau_2 \, G_{-E}(t_1^\star, \tau_2) J_E(\tau_2),
\end{aligned}
\tag{3.9b}
$$

$$
\begin{aligned}
Q_E(\tau_1) = q_E(\tau_1) & + \int_{t_0}^{t_f} dt_2 \, G_{E+}(\tau_1, t_2) J_+(t_2) - \int_{t_0}^{t_f} dt_2^\star \, G_{E-}(\tau_1, t_2^\star) J_-(t_2^\star) \\
& + i \int_{t_0}^{t_f} d\tau_2 \, G_{EE}(\tau_1, \tau_2) J_E(\tau_2),
\end{aligned}
\tag{3.9c}
$$

note that we include off diagonal terms. This is because the boundary conditions mix the different kind of fields therefore we expect some mixing in the propagator as well. The factors in front of the integrals are mostly conventional but they help match the factors in the integrals for the source terms. The $\star$ on the $t$s are just a convenience to remind which arguments belong to the forwards and backwards time segments.

The propagators need to obey the following equations:

$$\left(-\frac{\partial^2}{\partial t_1^2} - m^2\right)G_{++}(t_1, t_2) = \delta(t_1 - t_2),$$
(3.10a)

$$\left(-\frac{\partial^2}{\partial t_1^{\star 2}} - m^2\right)G_{-+}(t_1^\star, t_2) = 0,$$
(3.10b)

$$\left(-\frac{\partial^2}{\partial \tau_1^2} + m^2\right)G_{E+}(\tau_1, t_2) = 0,$$
(3.10c)

$$\left(-\frac{\partial^2}{\partial t_1^2} - m^2\right)G_{+-}(t_1, t_2^\star) = 0,$$
(3.10d)

$$\left(-\frac{\partial^2}{\partial t_1^{\star 2}} - m^2\right)G_{--}(t_1^\star, t_2^\star) = -\delta(t_1^\star - t_2^\star),$$
(3.10e)

$$\left(-\frac{\partial^2}{\partial \tau_1^2} + m^2\right)G_{E-}(\tau_1, t_2^\star) = 0,$$
(3.10f)

$$\left(-\frac{\partial^2}{\partial t_1^2} - m^2\right)G_{+E}(t_1, \tau_2) = 0,$$
(3.10g)

$$\left(-\frac{\partial^2}{\partial t_1^{\star 2}} - m^2\right)G_{-E}(t_1^\star, \tau_2) = 0,$$
(3.10h)

$$\left(-\frac{\partial^2}{\partial \tau_1^2} + m^2\right)G_{EE}(\tau_1, \tau_2) = -i\delta(\tau_1 - \tau_2),$$
(3.10i)

with boundary conditions coming from the field boundary conditions:

$$G_{++}(t_f, t_2) = G_{-+}(t_f, t_2), \qquad \left.\frac{\partial G_{++}(t_1, t_2)}{\partial t_1}\right|_{t_1=t_f} = \left.\frac{\partial G_{-+}(t_1^\star, t_2)}{\partial t_1^\star}\right|_{t_1^\star=t_f}, \qquad \text{(3.11a)}$$

$$G_{-+}(t_0, t_2) = G_{E+}(0, t_2), \qquad \left.\frac{\partial G_{-+}(t_1^\star, t_2)}{\partial t_1^\star}\right|_{t_1^\star=t_0} = i\left.\frac{\partial G_{E+}(\tau_1, t_2)}{\partial \tau_1}\right|_{\tau_1=0}, \qquad \text{(3.11b)}$$

$$G_{E+}(\beta, t_2) = G_{++}(t_0, t_2), \qquad i\left.\frac{\partial G_{E+}(\tau_1, t_2)}{\partial \tau_1}\right|_{\tau_1=\beta} = \left.\frac{\partial G_{++}(t_1, t_2)}{\partial t_1}\right|_{t_1=t_0}, \qquad \text{(3.11c)}$$

$$G_{+-}(t_f, t_2^\star) = G_{--}(t_f, t_2^\star), \qquad \left.\frac{\partial G_{+-}(t_1, t_2^\star)}{\partial t_1}\right|_{t_1=t_f} = \left.\frac{\partial G_{--}(t_1^\star, t_2^\star)}{\partial t_1^\star}\right|_{t_1^\star=t_f}, \qquad \text{(3.11d)}$$

$$G_{--}(t_0, t_2^\star) = G_{E-}(0, t_2^\star), \qquad \left.\frac{\partial G_{--}(t_1^\star, t_2^\star)}{\partial t_1^\star}\right|_{t_1^\star=t_0} = i\left.\frac{\partial G_{E-}(\tau_1, t_2^\star)}{\partial \tau_1}\right|_{\tau_1=0}, \qquad \text{(3.11e)}$$

$$G_{E-}(\beta, t_2^\star) = G_{+-}(t_0, t_2^\star), \qquad i\left.\frac{\partial G_{E-}(\tau_1, t_2)}{\partial \tau_1}\right|_{\tau_1=\beta} = \left.\frac{\partial G_{+-}(t_1, t_2^\star)}{\partial t_1}\right|_{t_1=t_0}, \qquad \text{(3.11f)}$$

$$G_{+E}(t_f, \tau_2) = G_{-E}(t_f, \tau_2), \qquad \left.\frac{\partial G_{+E}(t_1, \tau_2)}{\partial t_1}\right|_{t_1=t_f} = \left.\frac{\partial G_{+E}(t_1, \tau_2)}{\partial t_1}\right|_{t_1=t_f}, \qquad \text{(3.11g)}$$

$$G_{-E}(t_0, \tau_2) = G_{EE}(0, t_2), \qquad \left.\frac{\partial G_{-E}(t_1^\star, \tau_2)}{\partial t_1^\star}\right|_{t_1^\star=t_0} = i\left.\frac{\partial G_{EE}(\tau_1, \tau_2)}{\partial \tau_1}\right|_{\tau_1=0}, \qquad \text{(3.11h)}$$

$$G_{EE}(\beta, \tau_2) = G_{+E}(t_0, \tau_2), \qquad i\left.\frac{\partial G_{EE}(\tau_1, \tau_2)}{\partial \tau_1}\right|_{\tau_1=\beta} = \left.\frac{\partial G_{+E}(t_1, \tau_2)}{\partial t_1}\right|_{t_1=t_0}, \qquad \text{(3.11i)}$$

so that $Q_\pm$ and $Q_E$ have vanishing boundary conditions. They are ordered them in this particular way to highlight that even though they are nine coupled equations they come in three cycles of three equations each. Also note that the boundary conditions are only imposed in the first argument, the only way the two arguments mix is via the delta functions in the diagonal components. There is a diagonal component in each set so all equations end up mixing the two arguments.

After these simplifications it is fairly straightforward to solve the equations to get:

$$G_{++}(t_1, t_2) = -\frac{i}{2m} \cos\left(m\left(t_1 - t_2 - \frac{i\beta}{2}\right)\right) \operatorname{csch}\left(\frac{m\beta}{2}\right)$$
$$-\frac{1}{m}\Theta(t_1 - t_2)\sin(m(t_1 - t_2)), \tag{3.12a}$$

$$G_{--}(t_1^\star, t_2^\star) = -\frac{i}{2m} \cos\left(m\left(t_1^\star - t_2^\star + \frac{i\beta}{2}\right)\right) \operatorname{csch}\left(\frac{m\beta}{2}\right)$$
$$+\frac{1}{m}\Theta(t_1^\star - t_2^\star)\sin\left(m(t_1^\star - t_2^\star)\right), \tag{3.12b}$$

$$G_{EE}(\tau_1, \tau_2) = -\frac{i}{2m} \cosh\left(m\left(\tau_1 - \tau_2 + \frac{\beta}{2}\right)\right) \operatorname{csch}\left(\frac{m\beta}{2}\right)$$
$$+\frac{1}{m}\Theta(\tau_1 - \tau_2)\sinh(m(\tau_1 - \tau_2)), \tag{3.12c}$$

$$G_{+-}(t_1, t_2^\star) = -\frac{i}{2m} \cos\left(m\left(t_1 - t_2^\star - \frac{i\beta}{2}\right)\right) \operatorname{csch}\left(\frac{m\beta}{2}\right) = G_{-+}(t_2^\star, t_1), \tag{3.12d}$$

$$G_{+E}(t_1, \tau_2) = -\frac{i}{2m} \cos\left(m\left(t_1 - t_0 + i\tau_2 - \frac{i\beta}{2}\right)\right) \operatorname{csch}\left(\frac{m\beta}{2}\right) = G_{E+}(\tau_2, t_1), \tag{3.12e}$$

$$G_{-E}(t_1^\star, \tau_2) = -\frac{i}{2m} \cos\left(m\left(t_1^\star - t_0 + i\tau_2 - \frac{i\beta}{2}\right)\right) \operatorname{csch}\left(\frac{m\beta}{2}\right) = G_{E-}(\tau_2, t_1^\star). \tag{3.12f}$$

Symmetrising the diagonal components and inserting $1 = \Theta(t_1 - t_2) + \Theta(t_2 - t_1)$ we get:

$$G_{++}^{\text{sym}}(t_1, t_2) = -\frac{i}{2m} \cos\left(m\left(|t_1 - t_2| + \frac{i\beta}{2}\right)\right) \operatorname{csch}\left(\frac{m\beta}{2}\right), \tag{3.13a}$$

$$G_{--}^{\text{sym}}(t_1^\star, t_2^\star) = -\frac{i}{2m} \cos\left(m\left(|t_1^\star - t_2^\star| - \frac{i\beta}{2}\right)\right) \operatorname{csch}\left(\frac{m\beta}{2}\right), \tag{3.13b}$$

$$G_{EE}^{\text{sym}}(\tau_1, \tau_2) = -\frac{i}{2m} \cosh\left(m\left(|\tau_1 - \tau_2| - \frac{\beta}{2}\right)\right) \operatorname{csch}\left(\frac{m\beta}{2}\right). \tag{3.13c}$$

We now have nine propagators which seem largely independent. Nevertheless, there are some symmetries that can be exploited to reduce the number of propagators we actually have to consider. This is accomplished by changing to the average-difference basis, also called the Keldysh basis [7–13, 16, 18, 20, 53, 55, 58].

We define,

$$J_{\text{ave}} = \frac{J_+ + J_-}{2}, \qquad J_{\text{dif}} = J_+ - J_-, \tag{3.14a}$$

$$q_{\text{ave}} = \frac{q_+ + q_-}{2}, \qquad q_{\text{dif}} = q_+ - q_-. \tag{3.14b}$$

Plugging this into the above and using the fact that

$$G_{++}^{\text{sym}}(t_1, t_2) + G_{--}^{\text{sym}}(t_1, t_2) = G_{+-}(t_1, t_2) + G_{-+}(t_1, t_2), \tag{3.15}$$

we get

$$Z = \exp\left\{ -\frac{\mathrm{i}}{2} \int \mathrm{d}t_1 \,\mathrm{d}t_2 \, J_{\mathrm{dif}}(t_1) G_{\mathrm{ave,ave}}(t_1, t_2) J_{\mathrm{dif}}(t_2) \right.$$

$$-\frac{\mathrm{i}}{2} \int \mathrm{d}t_1 \,\mathrm{d}t_2 \, J_{\mathrm{ave}}(t_1) G_{\mathrm{dif,ave}}(t_1, t_2) J_{\mathrm{dif}}(t_2)$$

$$-\frac{\mathrm{i}}{2} \int \mathrm{d}t_1 \,\mathrm{d}t_2 \, J_{\mathrm{dif}}(t_1) G_{\mathrm{ave,dif}}(t_1, t_2) J_{\mathrm{ave}}(t_2)$$

$$+\frac{1}{2} \int \mathrm{d}t_1 \,\mathrm{d}\tau_2 \, J_{\mathrm{dif}}(t_1) G_{\mathrm{ave,E}}(t_1, \tau_2) J_{\mathrm{E}}(\tau_2)$$

$$+\frac{1}{2} \int \mathrm{d}\tau_1 \,\mathrm{d}t_2 \, J_{\mathrm{E}}(\tau_1) G_{\mathrm{E,ave}}(\tau_1, t_2) J_{\mathrm{dif}}(t_2)$$

$$\left. +\frac{\mathrm{i}}{2} \int \mathrm{d}\tau_1 \,\mathrm{d}\tau_2 \, J_{\mathrm{E}}(\tau_1) G_{\mathrm{E,E}}(\tau_1, \tau_2) J_{\mathrm{E}}(\tau_2) \right\}, \tag{3.16}$$

where

$$G_{\mathrm{ave,ave}}(t_1, t_2) = -\frac{\mathrm{i}}{2m} \cos(m(t_1 - t_2)) \coth\left(\frac{m\beta}{2}\right), \tag{3.17a}$$

$$G_{\mathrm{dif,ave}}(t_1, t_2) = \frac{1}{m} \sin(m(t_1 - t_2)) \Theta(t_2 - t_1) = G_{\mathrm{ave,dif}}(t_2, t_1), \tag{3.17b}$$

$$G_{\mathrm{ave,E}}(t_1, \tau_2) = G_{+,E}(t_1, \tau_2) = G_{-E}(t_1, \tau_2) = G_{\mathrm{E,ave}}(\tau_2, t_1). \tag{3.17c}$$

Note that the $J_{\mathrm{ave}}J_{\mathrm{ave}}$ and the $J_{\mathrm{ave}}J_E$ terms vanish identically. Also note that we have labelled the propagators so that any 'dif' label is together with a $J_{\mathrm{ave}}$ and vice-versa, this is on purpose because

$$J_+ q_+ - J_- q_- = J_{\mathrm{ave}} q_{\mathrm{dif}} + J_{\mathrm{dif}} q_{\mathrm{ave}}. \tag{3.18}$$

With this convention the 'dif' and 'ave' labels on diagrams will coincide with that will appear in correlators as functions of fields and with what appears in the potential.

## 3.3 Feynman rules in the average-difference basis

To deduce the Feynman rules we have to be careful with factors of i and $-1$ due to the mixing between real and imaginary fields, in App. B we present the derivation, in the main text we will just present the result.

For the average-difference basis in particular, since in $G_{\mathrm{dif,ave}}(t_1, t_2)$ we know that $t_2 > t_1$ we will draw an arrow from 'dif' to 'ave'. This flow implied by the arrows is usually called 'causal flow' because it tells you the direction of time. It is straightforward to see we cannot have a closed 'causal' loop, because we would have products of Heaviside-$\Theta$s that would always vanish. The other propagators do not have any causal connections but for ease of visibility there will always be arrows pointing towards a 'ave' end and legs that connect with Euclidean times will be dashed. In summary, here's the notation we shall use:

$$t_1 \longleftrightarrow t_2 \; = \mathrm{i}G_{\mathrm{ave,ave}}(t_1, t_2), \tag{3.19a}$$

$$t_1 \longrightarrow t_2 \; = \mathrm{i}G_{\mathrm{dif,ave}}(t_1, t_2), \tag{3.19b}$$

$$t_1 \dashleftarrow \tau_2 \; = \mathrm{i}G_{\mathrm{ave,E}}(t_1, \tau_2), \tag{3.19c}$$

$$\tau_1 \text{------------} \tau_2 = iG_{E,E}(\tau_1, \tau_2). \tag{3.19d}$$

In terms of vertices, there are three kinds. We have a quartic Euclidean vertex, and two Lorentzian ones. Since

$$\frac{1}{4!}q^4 - \frac{1}{4!}q'^4 = \frac{q_{ave}^3}{3!}q_{dif} + \frac{1}{4}q_{ave}\frac{q_{dif}^3}{3!}, \tag{3.20}$$

there is one Lorentzian vertex with three 'ave' and one 'dif' and another with three 'dif' and one 'ave'. Because there are only three identical legs in these vertices, the vertex with three 'dif' comes with an additional factor of $\frac{1}{4}$. In summary, we have:

$$= -i\lambda \int dt, \tag{3.21a}$$

$$= -i\frac{\lambda}{4} \int dt, \tag{3.21b}$$

$$= -\lambda \int d\tau, \tag{3.21c}$$

where in the last rule the dashed external legs may also have arrows if they come form a $G_{ave,E}$.

In higher dimensions all of the propagators also carry a momentum label. We should proceed exactly as in ordinary Feynman rules, we impose momentum conservation along propagators and vertices, and we integrate over loop momenta. Throughout the paper we shall drop overall momentum conserving Dirac-$\delta$s for ease of notation.

# 4 One-Loop symmetric propagator

We now compute the symmetric 2-point function $\langle\{\phi(x_1), \phi(x_2)\}\rangle$. In the average-difference basis it becomes:

$$\begin{aligned}
\langle\{\phi(x_1), \phi(x_2)\}\rangle &= \langle\phi_+(x_1)\phi_-(x_2) + \phi_-(x_1)\phi_+(x_2)\rangle \\
&= \left\langle \left(\phi_{ave}(x_1) + \frac{\phi_{dif}(x_2)}{2}\right)\left(\phi_{ave}(x_2) - \frac{\phi_{dif}(x_1)}{2}\right) \right. \\
&\quad \left. + \left(\phi_{ave}(x_1) - \frac{\phi_{dif}(x_2)}{2}\right)\left(\phi_{ave}(x_2) + \frac{\phi_{dif}(x_1)}{2}\right) \right\rangle \\
&= \left\langle 2\phi_{ave}(x_1)\phi_{ave}(x_2) - \frac{1}{2}\phi_{dif}(x_1)\phi_{dif}(x_2) \right\rangle, \tag{4.1}
\end{aligned}$$

where in the first line we have forced the ordering by placing one of the field operators in the forward moving segment (which appears first in the time contour) and the other on the backwards moving segment (which appears later in the contour). Also note that the last term in Eq. (4.1) vanishes (at least up to one-loop).

The diagrams that contribute to the symmetric 2-point function at 1-loop level are:

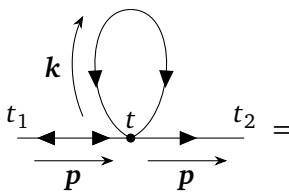

$$= -\frac{i\lambda}{2} \int_{t_0}^{t_f} dt \int \frac{d^d k}{(2\pi)^d} iG_{\text{ave,ave}}(\boldsymbol{p}, t_1, t) iG_{\text{ave,ave}}(\boldsymbol{k}, t, t) iG_{\text{dif,ave}}(\boldsymbol{p}, t, t_2)$$

$$= -\frac{\lambda}{2} \int_{t_0}^{t_f} dt \int \frac{d^d k}{(2\pi)^d} \frac{-i}{2E_{\boldsymbol{p}}} \cos[E_{\boldsymbol{p}}(t_1 - t)] \coth\left(\frac{E_{\boldsymbol{p}}\beta}{2}\right) \frac{-i}{2E_{\boldsymbol{k}}} \coth\left(\frac{E_{\boldsymbol{k}}\beta}{2}\right)$$

$$\times \frac{1}{E_{\boldsymbol{p}}} \sin[E_{\boldsymbol{p}}(t - t_2)]\Theta(t_2 - t)$$

$$= \frac{\lambda}{32 E_{\boldsymbol{p}}^3} \coth\left(\frac{E_{\boldsymbol{p}}\beta}{2}\right) \int \frac{d^d k}{(2\pi)^d} \frac{\coth\left(\frac{E_{\boldsymbol{k}}\beta}{2}\right)}{E_{\boldsymbol{k}}}$$

$$\times \left( \cos[E_{\boldsymbol{p}}(t_1 + t_2 - 2t_0)] - \cos[E_{\boldsymbol{p}}(t_1 - t_2)] + 2E_{\boldsymbol{p}}(t_2 - t_0)\sin[E_{\boldsymbol{p}}(t_1 - t_2)] \right), \qquad (4.2)$$

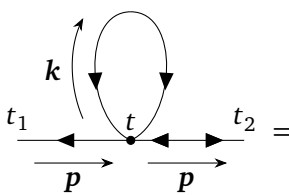

$$= -\frac{i\lambda}{2} \int_{t_0}^{t_f} dt \int \frac{d^d k}{(2\pi)^d} iG_{\text{ave,dif}}(\boldsymbol{p}, t_1, t) iG_{\text{ave,ave}}(\boldsymbol{k}, t, t) iG_{\text{ave,ave}}(\boldsymbol{p}, t, t_2)$$

$$= -\frac{\lambda}{2} \int_{t_0}^{t_f} dt \int \frac{d^d k}{(2\pi)^d} \frac{1}{E_{\boldsymbol{p}}} \sin[E_{\boldsymbol{p}}(t - t_1)]\Theta(t_1 - t) \frac{-i}{2E_{\boldsymbol{k}}} \coth\left(\frac{E_{\boldsymbol{k}}\beta}{2}\right)$$

$$\times \frac{-i}{2E_{\boldsymbol{p}}} \cos[E_{\boldsymbol{p}}(t - t_2)] \coth\left(\frac{E_{\boldsymbol{p}}\beta}{2}\right)$$

$$= \frac{\lambda}{32 E_{\boldsymbol{p}}^3} \coth\left(\frac{E_{\boldsymbol{p}}\beta}{2}\right) \int \frac{d^d k}{(2\pi)^d} \frac{\coth\left(\frac{E_{\boldsymbol{k}}\beta}{2}\right)}{E_{\boldsymbol{k}}}$$

$$\times \left( \cos[E_{\boldsymbol{p}}(t_1 + t_2 - 2t_0)] - \cos[E_{\boldsymbol{p}}(t_1 - t_2)] - 2E_{\boldsymbol{p}}(t_1 - t_0)\sin[E_{\boldsymbol{p}}(t_1 - t_2)] \right), \qquad (4.3)$$

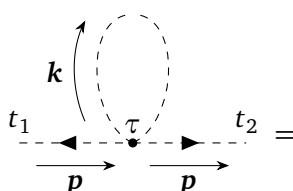

$$= -\frac{\lambda}{2}\int_0^\beta d\tau \int \frac{d^d k}{(2\pi)^d} iG_{\text{ave,E}}(\boldsymbol{p}, t_1, \tau) iG_{\text{E,E}}(\boldsymbol{k}, \tau, \tau) iG_{\text{E,ave}}(\boldsymbol{p}, \tau, t_2)$$

$$= \frac{i\lambda}{2}\int_0^\beta d\tau \int \frac{d^d k}{(2\pi)^d} \frac{-i}{2E_{\boldsymbol{p}}}\cos\left[E_{\boldsymbol{p}}\left(t_1 - t_0 + i\tau - i\frac{\beta}{2}\right)\right]\text{csch}\left(\frac{E_{\boldsymbol{p}}\beta}{2}\right)\frac{-i}{2E_{\boldsymbol{k}}}\cosh\left(-\frac{E_{\boldsymbol{k}}\beta}{2}\right)\text{csch}\left(\frac{E_{\boldsymbol{k}}\beta}{2}\right)$$

$$\times \frac{-i}{2E_{\boldsymbol{p}}}\cos\left[E_{\boldsymbol{p}}\left(t_2 - t_0 + i\tau - i\frac{\beta}{2}\right)\right]\text{csch}\left(\frac{E_{\boldsymbol{p}}\beta}{2}\right)$$

$$= -\frac{\lambda}{32E_{\boldsymbol{p}}^3}\left(\text{csch}^2\left(\frac{E_{\boldsymbol{p}}\beta}{2}\right)E_{\boldsymbol{p}}\beta\cos\left[E_{\boldsymbol{p}}(t_1 - t_2)\right] + 2\coth\left(\frac{E_{\boldsymbol{p}}\beta}{2}\right)\cos\left[E_{\boldsymbol{p}}(t_1 + t_2 - 2t_0)\right]\right)$$

$$\times \int \frac{d^d k}{(2\pi)^d}\frac{\coth\left(\frac{E_{\boldsymbol{k}}\beta}{2}\right)}{E_{\boldsymbol{k}}}, \tag{4.4}$$

notice how in this diagram we have contributions which do not depend on $t_0$. Therefore, even for initial conditions set in the infinite past you need to include these cross terms.

Adding it all up we get:

$$\left\langle\left\{\tilde{\phi}(\boldsymbol{p}, t_1), \tilde{\phi}(-\boldsymbol{p}, t_2)\right\}\right\rangle_{\text{1-loop}} = 2\left\langle\tilde{\phi}_{\text{ave}}(\boldsymbol{p}, t_1)\tilde{\phi}_{\text{ave}}(-\boldsymbol{p}, t_2)\right\rangle_{\text{1-loop}}$$

$$= -\frac{\lambda}{16E_{\boldsymbol{p}}^3}\int \frac{d^d k}{(2\pi)^d}\frac{\coth\left(\frac{E_{\boldsymbol{k}}\beta}{2}\right)}{E_{\boldsymbol{k}}}\left(2\coth\left(\frac{E_{\boldsymbol{p}}\beta}{2}\right)\left(E_{\boldsymbol{p}}(t_1 - t_2)\sin\left[E_{\boldsymbol{p}}(t_1 - t_2)\right]\right.\right.$$

$$\left.\left. + \cos\left[E_{\boldsymbol{p}}(t_1 - t_2)\right]\right) + \text{csch}^2\left(\frac{E_{\boldsymbol{p}}\beta}{2}\right)E_{\boldsymbol{p}}\beta\cos\left[E_{\boldsymbol{p}}(t_1 - t_2)\right]\right). \tag{4.5}$$

Note that the $t_0$ dependence cancelled between the three diagrams as is to be expected from the time-translation invariance of the thermal state.

We still need to add the counterterms. Usually we need to resum the series to consider 1PI graphs [1,3,4,57], but this is much harder in this formalism, so what we shall do instead is to make $m^2 \to m^2 + \delta m^2$ in the tree-level answer and expand in powers of $\delta m^2$. The idea is that $\delta m^2$ is linear in $\lambda$. This is actually a bit closer to the spirit of renormalisation, we are figuring out what is the function $m^2(\lambda, \Lambda)$ that we need to put in the action so that $m^2$ corresponds to the physical measured mass (squared) and then expanding in powers of $\lambda$ ($\Lambda$ is the cutoff, we'll be mostly agnostic about how exactly we are regulating the theory). We then get:

$$-\frac{i\coth\left(\frac{1}{2}\beta\sqrt{m^2 + \delta m^2 + \boldsymbol{p}^2}\right)\cos\left[(t_1 - t_2)\sqrt{m^2 + \delta m^2 + \boldsymbol{p}^2}\right]}{2\sqrt{m^2 + \delta m^2 + \boldsymbol{p}^2}}$$

$$= -\frac{i}{2E_{\boldsymbol{p}}}\cos\left(E_{\boldsymbol{p}}(t_1 - t_2)\right)\coth\left(\frac{E_{\boldsymbol{p}}\beta}{2}\right)$$

$$+ \left(2\coth\left(\frac{E_{\boldsymbol{p}}\beta}{2}\right)\left(E_{\boldsymbol{p}}(t_1 - t_2)\sin\left[E_{\boldsymbol{p}}(t_1 - t_2)\right] + \cos\left[E_{\boldsymbol{p}}(t_1 - t_2)\right]\right)\right.$$

$$\left. + \text{csch}^2\left(\frac{E_{\boldsymbol{p}}\beta}{2}\right)E_{\boldsymbol{p}}\beta\cos\left[E_{\boldsymbol{p}}(t_1 - t_2)\right]\right)\frac{i\delta m^2}{8E_{\boldsymbol{p}}^3} + O(\delta m^2)^2. \tag{4.6}$$

The contribution to the symmetric 2-point function at $O(\lambda)$ is then:

$$\left\langle \left\{ \tilde{\phi}(\boldsymbol{p}, t_1), \tilde{\phi}(-\boldsymbol{p}, t_2) \right\} \right\rangle_{\delta m^2} = -\frac{\delta m^2}{4E_{\boldsymbol{p}}^3} \left( 2 \coth\left( \frac{E_{\boldsymbol{p}}\beta}{2} \right) \left( E_{\boldsymbol{p}}(t_1 - t_2) \sin\left[ E_{\boldsymbol{p}}(t_1 - t_2) \right] \right. \right.$$

$$\left. + \cos\left[ E_{\boldsymbol{p}}(t_1 - t_2) \right] \right) + \operatorname{csch}^2\left( \frac{E_{\boldsymbol{p}}\beta}{2} \right) E_{\boldsymbol{p}}\beta \cos\left[ E_{\boldsymbol{p}}(t_1 - t_2) \right] \Bigg). \quad (4.7)$$

Similarly, there is also the question of field renormalisation. In the same vein as above, what we need to do is insert a $Z(\lambda, \Lambda)$ as a coefficient to the kinetic term, expand in powers of $\lambda$ and figure out what is the physical normalisation. This avoids dealing with diagrams with time derivatives. Naively it seems like we need to solve the equations once again, however, by looking at the derivation of Eq. (3.10) we see that adding $Z$ would correspond to multiplying the $\frac{\partial^2}{\partial t^2}$ terms by $Z$. However, if we define $m'^2 = \frac{m^2}{Z}$ and $G' = ZG$ then $G'$ solves the same equation as if we had no field renormalisation since the boundary conditions don't depend on the normalisation of $G$. Therefore, we have:

$$G'_{\text{ave,ave}}(\boldsymbol{p}, t_1, t_2) = -\frac{i \coth\left( \frac{1}{2}\beta \sqrt{m'^2 + \boldsymbol{p}^2} \right) \cos\left[ (t_1 - t_2) \sqrt{m'^2 + \boldsymbol{p}^2} \right]}{2\sqrt{m'^2 + \boldsymbol{p}^2}} \Longleftrightarrow$$

$$G_{\text{ave,ave}}(\boldsymbol{p}, t_1, t_2) = -\frac{i \coth\left( \frac{1}{2}\beta \sqrt{\frac{m^2}{Z} + \boldsymbol{p}^2} \right) \cos\left[ (t_1 - t_2) \sqrt{\frac{m^2}{Z} + \boldsymbol{p}^2} \right]}{2Z\sqrt{\frac{m^2}{Z} + \boldsymbol{p}^2}}. \quad (4.8)$$

Now expanding in powers of $\lambda$ as $Z = 1 + \delta Z$ we get:

$$= -\frac{i}{2E_{\boldsymbol{p}}} \cos\left( E_{\boldsymbol{p}}(t_1 - t_2) \right) \coth\left( \frac{E_{\boldsymbol{p}}\beta}{2} \right)$$

$$- \left( 2 \coth\left( \frac{E_{\boldsymbol{p}}\beta}{2} \right) \left( E_{\boldsymbol{p}}(t_1 - t_2) \sin\left[ E_{\boldsymbol{p}}(t_1 - t_2) \right] \right. \right.$$

$$+ \left( 1 - \frac{2E_{\boldsymbol{p}}^2}{m^2} \right) \cos\left[ E_{\boldsymbol{p}}(t_1 - t_2) \right] \bigg)$$

$$+ \operatorname{csch}^2\left( \frac{E_{\boldsymbol{p}}\beta}{2} \right) E_{\boldsymbol{p}}\beta \cos\left[ E_{\boldsymbol{p}}(t_1 - t_2) \right] \Bigg) \frac{i m^2 \delta Z}{8E_{\boldsymbol{p}}^3} + O(\delta Z)^2, \quad (4.9)$$

which is very similar to the mass counterterm, except it has an additional term.

The full 1-loop contribution to the symmetric 2-point function including counterterms is:

$$\left\langle \left\{ \tilde{\phi}(\boldsymbol{p}, t_1), \tilde{\phi}(-\boldsymbol{p}, t_2) \right\} \right\rangle_{\text{1-loop+c.t.}}$$

$$= -\frac{\lambda I_\beta(\Lambda) + 4\delta m^2 - 4m^2 \delta Z}{16E_{\boldsymbol{p}}^3} \left( 2 \coth\left( \frac{E_{\boldsymbol{p}}\beta}{2} \right) \left( E_{\boldsymbol{p}}(t_1 - t_2) \sin\left[ E_{\boldsymbol{p}}(t_1 - t_2) \right] \right. \right.$$

$$+ \cos\left[ E_{\boldsymbol{p}}(t_1 - t_2) \right] \bigg) + \operatorname{csch}^2\left( \frac{E_{\boldsymbol{p}}\beta}{2} \right) E_{\boldsymbol{p}}\beta \cos\left[ E_{\boldsymbol{p}}(t_1 - t_2) \right] \bigg) - \frac{\delta Z}{E_{\boldsymbol{p}}} \cos\left[ E_{\boldsymbol{p}}(t_1 - t_2) \right], \quad (4.10)$$

where

$$I_\beta(\Lambda) = \int \frac{d^d k}{(2\pi)^d} \frac{\coth\left( \frac{E_{\boldsymbol{k}}\beta}{2} \right)}{E_{\boldsymbol{k}}}, \quad (4.11)$$

and the integral is assumed to be regulated in some way.

## 4.1 Choice of counterterms

In order to choose an appropriate $\delta m^2$ and $\delta Z$ we need some physical definition of mass and field renormalisation. Given these are parameters in the action/Hamiltonian we do not expect them to depend on the temperature. For example, if the mass is defined as the energy gap in the spectrum, this will be a feature of the Hamiltonian rather than of the initial state we put our system in. This means we should take the zero temperature limit and then use the usual Källén-Lehmann spectral representation [1, 3] to get an appropriate definition of mass and field renormalisation.

The $\beta \to \infty$ limit of the above reads

$$
\begin{aligned}
\left\langle \left\{ \tilde{\phi}(\boldsymbol{p}, t_1), \tilde{\phi}(-\boldsymbol{p}, t_2) \right\} \right\rangle_{\text{1-loop+c.t.}}^{\beta \to \infty} &= \langle \Omega | \left\{ \tilde{\phi}(\boldsymbol{p}, t_1), \tilde{\phi}(-\boldsymbol{p}, t_2) \right\} | \Omega \rangle_{\text{1-loop+c.t.}} \\
&= -\frac{\lambda I_\infty(\Lambda) + 4\delta m^2 - 4m^2 \delta Z}{8E_{\boldsymbol{p}}^3} \Big( E_{\boldsymbol{p}}(t_1 - t_2) \sin\left[ E_{\boldsymbol{p}}(t_1 - t_2) \right] + \cos\left[ E_{\boldsymbol{p}}(t_1 - t_2) \right] \Big) \\
&\quad - \frac{\delta Z}{E_{\boldsymbol{p}}} \cos\left[ E_{\boldsymbol{p}}(t_1 - t_2) \right],
\end{aligned}
\tag{4.12}
$$

where

$$
I_\infty(\Lambda) = \int \frac{\mathrm{d}^d k}{(2\pi)^d} \frac{1}{E_{\boldsymbol{k}}},
\tag{4.13}
$$

and $|\Omega\rangle$ is defined as the ground state of the Hamiltonian (in principle at time $t_0$). In the limit $\beta \to \infty$ this is the only state that contributes.

By running the usual arguments for the Källén-Lehmann spectral representation [1,3] but for the symmetric 2-point function we get

$$
\langle \Omega | \left\{ \tilde{\phi}(\boldsymbol{p}, t_1), \tilde{\phi}(-\boldsymbol{p}, t_2) \right\} | \Omega \rangle = \int_0^\infty \frac{\mathrm{d}M^2}{2\pi} \frac{\rho(M^2)}{E_{\boldsymbol{p}}} \cos\left[ E_{\boldsymbol{p}}(t_1 - t_2) \right],
\tag{4.14}
$$

by setting $\boldsymbol{p} = \boldsymbol{0}$, $t_2 = 0$, and $t_1 = t$ to simplify our calculations ($\rho$ cannot depend on any of these variables by construction) it is straightforward to get

$$
\begin{aligned}
\rho(M^2) &= \left( 1 - \frac{\lambda I_\infty(\Lambda) + 4\delta m^2 - 4m^2 \delta Z}{8m^2} - \delta Z \right) 2\pi \delta(M^2 - m^2) \\
&\quad + \left( \frac{\lambda}{4} I_\infty(\Lambda) + \delta m^2 - m^2 \delta Z \right) 2\pi \frac{\partial}{\partial M^2} \left( \delta(M^2 - m^2) \right).
\end{aligned}
\tag{4.15}
$$

This seems like a bit of a weird behaviour since we get a delta function at $m^2$ but we also get a derivative of a delta function, which is more singular than would be expected. However, this is just an artefact of our perturbative expansion. In fact, this expression is equivalent to shifting the pole by an amount

$$
\Delta = \frac{\lambda}{4} I_\infty(\Lambda) + 2\delta m^2 - 2m^2 \delta Z,
\tag{4.16}
$$

that is, we can also write $\rho(M^2)$ as

$$
\rho(M^2) = \left( 1 - \frac{\Delta}{2m^2} - \frac{\delta Z}{2} \right) 2\pi \delta(M^2 - m^2 + \Delta),
\tag{4.17}
$$

and obtain the previous answer by expanding in powers of $\lambda$, $\delta m^2$, and $\delta Z$.

Our physical renormalisation conditions (choosing $m^2$ to be our physical mass) are that the pole is at $m^2$ and that the coefficient in front is 1. Solving for the counterterms we get:

$$\delta m^2 = -\frac{\lambda}{4} I_\infty(\Lambda), \tag{4.18a}$$

$$\delta Z = 0. \tag{4.18b}$$

The end result is then:

$$
\begin{aligned}
&\left\langle \left\{ \tilde{\phi}(\boldsymbol{p}, t_1), \tilde{\phi}(-\boldsymbol{p}, t_2) \right\} \right\rangle_{\text{1-loop+c.t.}} \\
&= -\frac{\lambda(I_\beta(\Lambda) - I_\infty(\Lambda))}{16 E_{\boldsymbol{p}}^3} \left( 2 \coth\left( \frac{E_{\boldsymbol{p}} \beta}{2} \right) \left( E_{\boldsymbol{p}}(t_1 - t_2) \sin\left[ E_{\boldsymbol{p}}(t_1 - t_2) \right] \right. \right. \\
&\qquad \left. \left. + \cos\left[ E_{\boldsymbol{p}}(t_1 - t_2) \right] \right) + \operatorname{csch}^2\left( \frac{E_{\boldsymbol{p}} \beta}{2} \right) E_{\boldsymbol{p}} \beta \cos\left[ E_{\boldsymbol{p}}(t_1 - t_2) \right] \right).
\end{aligned}
\tag{4.19}
$$

Note that the integral

$$\int \frac{\mathrm{d}^d k}{(2\pi)^d} \frac{\coth\left( \frac{E_{\boldsymbol{k}} \beta}{2} \right) - 1}{E_{\boldsymbol{k}}}, \tag{4.20}$$

is convergent even without a cutoff. With a finite cutoff it depends on the cutoff but that dependence is negligible if the cutoff is far above any scales of interest. This behaviour is exactly what is expected of a field theory at finite temperature [5–8, 10, 11, 13, 18, 21, 22, 25, 59, 60].

The final answer does not contain any terms proportional to $t_1 + t_2$ therefore there are no secular effects. However, there is still a temporal IR growth from the term proportional to $(t_1 - t_2)$. This does not affect the energy-momentum tensor (as it vanishes in the coincidence limit) but it means that naive perturbation theory is inadequate if the temporal separation is too large. However, this effect is easy to resum.

First note that if instead we chose a temperature dependent counterterm:

$$\delta m^2 = -\frac{\lambda}{4} I_\beta, \tag{4.21}$$

the mass parameter would not correspond to the physical mass as it won't be the energy gap in the spectrum, but the secular effect won't be there. It is also not very physical to have terms in the Hamiltonian that depend on the choice of initial conditions.[3] However, this tells us how to resum these terms.

Then note that the physical choice of counterterm means that the relation between the physical mass $m_{\text{phys}}^2$ and the mass parameter in the Lagrangian $m_{\text{Lagrangian}}^2$ is

$$m_{\text{Lagrangian}}^2 = m_{\text{phys}}^2 - \frac{\lambda}{4} I_\infty, \tag{4.22}$$

where $m_{\text{phys}}$ is independent of the regulator.

All in all this suggests that if we insert as a mass parameter in the propagators:

$$m_{\text{prop}}^2 = m_{\text{Lagrangian}}^2 + \frac{\lambda}{4} I_\beta = m_{\text{phys}}^2 + \frac{\lambda}{4}(I_\beta - I_\infty), \tag{4.23}$$

---

[3]The author thanks Stefan Hollands for pointing this out.

then we rescue perturbation theory at large temporal separations. Note that we are not inserting this in the Lagrangian, the claim is that the contribution from these diagrams could be resumed by using this modified propagator. This agrees with what is found in the literature for the thermal mass shift [5–8, 10, 11, 13, 18, 21, 22, 25, 59, 60].

Had we taken the naive approach and not considered the $G_{\text{ave,E}}$ cross terms we would have found several issues. Firstly, we would find that the final answer depends on $t_0$. This is to be expected, by disregarding these terms we are essentially taking $\rho = \exp(-\beta H_0)$ as our initial state, where $H_0$ is the free part of the Hamiltonian. Given the free Hamiltonian does not commute with the full Hamiltonian we ought to expect time dependence. However, this time dependence is not ameliorated by taking the limit $t_0 \to -\infty$ as the dependence is oscillatory rather than decaying. We could perhaps take the limit in such a way to turn those oscillations into damping [23, 50] however, we would then not recover the final term that arises from the cross terms which puts this method into question.

However, there is some evidence that in some sense $\rho = \exp(-\beta H_0)$ is 'close enough' to the desired state. Had we only included the $2 \times 2$ propagators and only included the counterterms in the interaction Hamiltonian rather than expanding the tree-level propagator as we did, we would obtain the correct IR resummation. This suggests there could be some dynamical effect which makes the two states agree once we fix their IR behaviour. Nevertheless, this claim relies on the fact this resummation would continue to agree at every loop level, which, to the knowledge of the author, has not been proven.

Further, we would have obtained a different answer depending on whether we do counterterms as usual (which corresponds to inserting them in the interaction Hamiltonian) or expanding the tree-level propagator (which corresponds to inserting them in the free Hamiltonian). This difference arises because the initial state depends on the free Hamiltonian but not the interaction Hamiltonian. This puts into question the mathematical consistency of the whole formalism.

To fully settle the debate, in the next section we explicitly calculate the equal-time 4-point function, checking whether or not it would be possible to get an agreement between the various approaches. Once more this is a very physical quantity to calculate as it is often the object of interest in, *e.g.* cosmological applications [29–42].

## 5  Tree-level equal-time 4-point function

We wish to calculate:

$$
\begin{aligned}
&\left\langle \tilde{\phi}(\boldsymbol{p}_1, t_f)\tilde{\phi}(\boldsymbol{p}_2, t_f)\tilde{\phi}(\boldsymbol{p}_3, t_f)\tilde{\phi}(\boldsymbol{p}_4, t_f)\right\rangle_\beta \\
&= \left\langle \tilde{\phi}_{\text{ave}}(\boldsymbol{p}_1, t_f)\tilde{\phi}_{\text{ave}}(\boldsymbol{p}_2, t_f)\tilde{\phi}_{\text{ave}}(\boldsymbol{p}_3, t_f)\tilde{\phi}_{\text{ave}}(\boldsymbol{p}_4, t_f)\right\rangle_\beta,
\end{aligned}
\tag{5.1}
$$

where, in the last line, we used the fact that the equal-time means we can use $\phi_\pm$ interchangeably and therefore we can use $\phi_{\text{ave}}$.

The diagrams that contribute are:

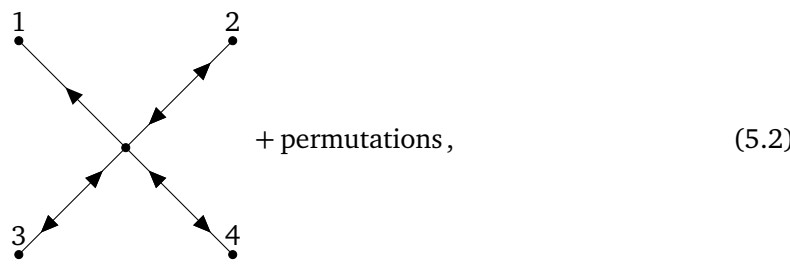

$$+ \text{ permutations}, \tag{5.2}$$

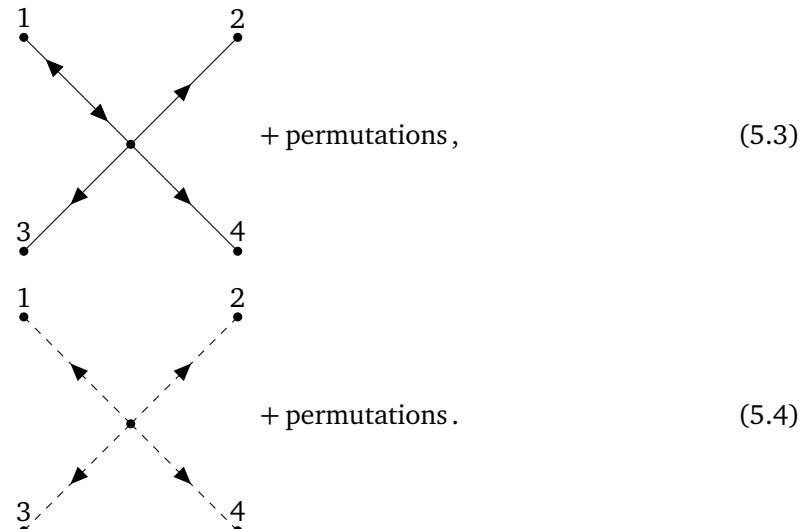

$$+ \text{permutations}, \qquad (5.3)$$

$$+ \text{permutations}. \qquad (5.4)$$

Let's choose $E_{\boldsymbol{p}_1} = E_{\boldsymbol{p}_2} = E_{\boldsymbol{p}_3} = E_{\boldsymbol{p}_4} = E$, or equivalently $|\boldsymbol{p}_1| = |\boldsymbol{p}_2| = |\boldsymbol{p}_3| = |\boldsymbol{p}_4|$ for simplicity, then

$$
\begin{aligned}
(5.2) &= -\mathrm{i}\lambda \int_{t_0}^{t_f} \mathrm{d}t \, \mathrm{i}G_{\mathrm{dif,ave}}(\boldsymbol{p}_1, t, t_f) \mathrm{i}G_{\mathrm{ave,ave}}(\boldsymbol{p}_2, t, t_f) \mathrm{i}G_{\mathrm{ave,ave}}(\boldsymbol{p}_3, t, t_f) \mathrm{i}G_{\mathrm{ave,ave}}(\boldsymbol{p}_4, t, t_f) \\
&\quad + \text{permutations} \\
&= -4\mathrm{i}\lambda \int_{t_0}^{t_f} \mathrm{d}t \, \frac{1}{E} \sin\big(E(t-t_f)\big) \underbrace{\Theta(t_f - t)}_{=1} \left( -\frac{\mathrm{i}}{2E} \cos\big(E(t-t_f)\big) \coth\left(\frac{E\beta}{2}\right) \right)^3 \\
&= -\frac{\lambda}{8E^5} \coth^3\left(\frac{E\beta}{2}\right) \big(1 - \cos^4(E\Delta t)\big),
\end{aligned} \qquad (5.5)
$$

where $\Delta t = t_f - t_0$.

$$
\begin{aligned}
(5.3) &= -\mathrm{i}\frac{\lambda}{4} \int_{t_0}^{t_f} \mathrm{d}t \, \mathrm{i}G_{\mathrm{ave,ave}}(\boldsymbol{p}_1, t, t_f) \mathrm{i}G_{\mathrm{dif,ave}}(\boldsymbol{p}_2, t, t_f) \mathrm{i}G_{\mathrm{dif,ave}}(\boldsymbol{p}_3, t, t_f) \mathrm{i}G_{\mathrm{dif,ave}}(\boldsymbol{p}_4, t, t_f) \\
&\quad + \text{permutations} \\
&= -\mathrm{i}\lambda \int_{t_0}^{t_f} \mathrm{d}t \, \frac{-\mathrm{i}}{2E} \cos\big(E(t-t_f)\big) \coth\left(\frac{E\beta}{2}\right) \left( \frac{1}{E} \sin\big(E(t-t_f)\big) \underbrace{\Theta(t_f - t)}_{=1} \right)^3 \\
&= \frac{\lambda}{8E^5} \coth\left(\frac{E\beta}{2}\right) \sin^4(E\Delta t),
\end{aligned} \qquad (5.6)
$$

$$
\begin{aligned}
(5.4) &= -\lambda \int_0^\beta \mathrm{d}\tau \, \mathrm{i}G_{\mathrm{ave,E}}(\boldsymbol{p}_1, t_f, \tau) \mathrm{i}G_{\mathrm{ave,E}}(\boldsymbol{p}_2, t_f, \tau) \mathrm{i}G_{\mathrm{ave,E}}(\boldsymbol{p}_3, t_f, \tau) \mathrm{i}G_{\mathrm{ave,E}}(\boldsymbol{p}_4, t_f, \tau) \\
&= -\lambda \int_0^\beta \mathrm{d}\tau \left( \frac{-\mathrm{i}}{2E} \cos\left( E\left( \Delta t + \mathrm{i}\tau - \mathrm{i}\frac{\beta}{2} \right) \right) \mathrm{csch}\left(\frac{E\beta}{2}\right) \right)^4 \\
&= -\frac{\lambda}{256E^5} \mathrm{csch}^4\left(\frac{E\beta}{2}\right) \big(6\beta E + 8\cos(2\Delta t)\sinh(\beta E) + \cos(4\Delta t)\sinh(2\beta E)\big).
\end{aligned} \qquad (5.7)
$$

Therefore, the total answer is

$$(5.2) + (5.3) + (5.4) = -\frac{\lambda}{256 E^5} \operatorname{csch}^4\left(\frac{\beta E}{2}\right)\left(6\beta E + 8\sinh(\beta E) + \sinh(2\beta E)\right). \qquad (5.8)$$

This end result is completely independent of time and fully agrees with an imaginary-time formalism calculation as it should. However, that time independence was once more only there due to the cross-terms. What is more, it is more accurate to say the real-time terms canceled the time dependence of the cross terms as the final answer comes purely from the cross terms. This is not recoverable from a modification of the quadratic components or the $2 \times 2$ propagator matrix. Further, it is now completely transparent that in no way the non-Gaussianities of the initial density matrix are damped or disappear at early times, in fact they are completely independent of time.

The only reasonable conclusion is that finite temperature quantum field theories are not free in the far past and that, if we wish to calculate higher point functions we must use the full $3 \times 3$ propagator matrix.

# 6 Conclusion

We conclude by contrasting this paper with what is found in the pre-existing body of literature.

The first main difference with the most common approaches is that, so far, we have not relied too heavily on transforming to Fourier space in time. This difference is mostly cosmetic but there are reasons behind the choice made in this paper.

Firstly, a priori, all our time variables exist in a compact time interval, either $[t_0, t_f]$ or $[0, \beta]$, therefore, naively, we cannot just Fouier transform.

However, we might wish to take a Fourier series instead. This is complicated by the fact none of our functions is periodic in these intervals individually. If we performed a Fourier series we would either ruin the boundary conditions for the value of the function or for its first derivative, we cannot keep both arbitrary.

Finally, one might want to leverage the fact the boundary conditions are joined in a loop as if the time variable was merely following a contour in the complex plane. This is perfectly legitimate in non-relativistic theories, which have first order equations of motion. However, for relativistic theories we run into a problem with matching the first derivatives. The issue is that, in order for this picture to work we would need to impose continuity of the first derivatives along the contour, which would actually mean imposing:

$$\dot{q}_+(t_f) = -\dot{q}_-(t_f), \qquad (6.1)$$

which does not cancel the boundary terms when integrating by parts.

These subtleties may be ameliorated if one takes the limits $t_0 \to -\infty$ and $t_f \to \infty$, but we do not wish to do at this stage to make sure we have not been sloppy with these limits. This is ultimately why we avoid going to temporal Fourier space and mostly do not speak in terms of the time contour.

On a related point, the average-difference basis is not the only basis which can provide simplifications. Namely, there is the retarded-advanced basis [16, 58] which takes advantage of the Kubo-Martin-Schwinger (KMS) relation:

$$G_{+-}(t_1 + \mathrm{i}\beta, t_2) = G_{-+}(t_1, t_2). \qquad (6.2)$$

However, this relates functions at different points in time, therefore it can only be easily used in Fourier space. For the reasons stated above we have avoided Fourier space and therefore not used the retarded-advanced basis. It is still important to note that there is even further structure in the propagators used in this paper.

The most important difference with the pre-existing literature is the treatment of the cross terms between the real and imaginary segments. In the vast majority of the literature they are simply disregarded [7,8,10,11,13–16,18,19,21,22,26]. There are several arguments that are used to justify not taking them into consideration, but, in essence, they boil down to taking the limit $t_0 \to -\infty$ and either just assuming the interactions decay at very early times [8,10] or changing the dynamics explicitly to forcibly turn off the interactions in the far past [49,50].

Up to an extent this is perfectly legitimate. After all we can use whatever Hamiltonian we wish and whatever initial conditions we wish. There is no mathematical or physical inconsistency with choosing the initial density matrix to be $\rho = \exp(-\beta H_0)$, where $H_0$ is the quadratic part of the Hamiltonian, or adding an exponential decay to the interaction Hamiltonian. The real question is whether or not this is accurately capturing thermal physics.

If one used the ad-hoc $\rho = \exp(-\beta H_0)$ the issue is that, in contrast with the full Gibbs state, it is not time independent, the free Hamiltonian does not commute with the interaction Hamiltonian. Therefore we would have to trust this state is in some sense 'close enough' to the true finite temperature state so that the difference in observables calculated with either state would small or decaying with time. In Sec. 4 and 5 we have explicitly compared these two methods and reached the conclusion no such mechanism appears to exist.

If one changed the Hamiltonian to turn off the interactions there are two ways in which we could test its accuracy at describing thermal physics. The first is by comparing with experimental results. The second is to take the limit in which this damping is removed, which is what is usually described as desired [49,50]. The issue with this last method is that the two limits may not commute. We may get different answers if we remove the damping before or after taking the $t_0 \to -\infty$. The calculations in Sec. 4 and 5 indeed demonstrate this will be the case.

There have also been some works in the past that tried to take the effect of the interactions into account [51,52,56,61]. Most notably, in the non-relativistic community these effects have been widely studied and it is even a matter of textbooks and reviews [60,62–64]. In this case it has even been argued that the $3 \times 3$ propagator matrix is equivalent to including an explicit coupling term to an external bath [64]. Nevertheless, the lessons from this case cannot be straightforwardly imported to relativistic theories. The main objection being that the propagator equations are first order in time which means time contour arguments are much more straightforward. The solutions are just distinct and there is no a priori reason that the arguments and proofs that work in that case can be extended to the relativistic case.

Another relatively known approach is that in [51,52] which attempts to give a prescription for how to modify the $2 \times 2$ propagators into giving the full answer. However, the arguments do not quite hold up to scrutiny as they do not correctly take into account the presence of internal Euclidean vertices. Indeed as the calculations in Sec. 5 demonstrate, no such reasoning can be true.

Finally, in [56] the role of the interactions is correctly taken into account and $t_0$ is held fixed until the very end by using a 2PI formalism. Unfortunately, none of the relativistic works that cite them correctly take interactions into account instead using the incorrect $2 \times 2$ propagator matrix. In [61] these effects are also taken into account but the technical points are mixed in with the disorder averaging, which complicates the interpretation.

All in all, despite the existence of some works which do take these effects into account misconceptions regarding the role of these interactions are overwhelming prevalent in the literature. The most popular textbooks and reviews, even recent ones, do not take these effects

into account. The author hopes this work can demonstrate in a simple manner the importance of the cross-terms and clear the confusion in the field.

## Acknowledgements

I would first like to thank Jorge Santos for giving an idea that would later evolve to become this project, for comments on an earlier draft of this paper, for many insightful discussions, and for extraordinary PhD supervision in such difficult times. I would like to thank Stefan Hollands, Joaquim António, Ben Freivogel, Jan Pieter van der Schaar, Victor Gorbenko, Dimitrii Diakonov, and Kirill Bazarov for very useful discussions and comments. I would also like to thank Owain Salter Fitz-Gibbons for many late night discussions which, while often fruitless, were nevertheless insightful. Finally I would like to thank the Cambridge Trust from my Vice Chancellor's award to support my studies.

## A  Derivative boundary conditions

From the propagator perspective, we have second order equations of motion so we need to impose boundary conditions on the first derivatives to get well posed equations. From the Lagrangian perspective, as we need to integrate by parts and the boundary term depends on first derivatives, we need some condition on the first derivatives to be able to manipulate the boundary terms.

However, from the canonical point of view, these boundary conditions appear because we have at some point inserted a complete set of states, but those states only depend on the field values, not its time derivatives. Furthermore, we have some intuition that we might need to consider fields which are not differentiable and deal with the discrete sum over time, rather than a continuous integral [4, 57]. How can we see the first derivative condition from this point of view?

The discrete derivative we introduce when doing the calculation is [57]

$$D(q_i) = \frac{q_{i+1} - q_i}{\epsilon}\,, \tag{A.1}$$

which obeys a modified product rule

$$D(q_i h_i) = D(q_i)h_{i+i} + q_i D(h_i)\,, \tag{A.2}$$

therefore the integration by parts is

$$\sum_{i=0}^{N-1} \epsilon D(q_i)D(q_i) = \sum_{i=0}^{N-1} \epsilon D(q_i D(q_i)) - \sum_{i=0}^{N-1} \epsilon D(D(q_i))q_{i+1}$$

$$= D(q_N)q_N - D(q_0)q_0 - \sum_{i=0}^{N-1} \epsilon D(D(q_i))q_{i+1}\,. \tag{A.3}$$

Notice how, to integrate by parts, we needed to introduce a $D(q_N)$ which depends on $q_{N+1}$ which seems to make this ill-defined, however, there is also a term that depends on the $q_{N+1}$ in the second derivative which cancels this contribution, hence the whole thing doesn't depend on $q_{N+1}$ and all is well. This then means that we are free to choose $q_{N+1}$ to be whatever we want as it cancels in the final answer. This is equivalent to a freedom in choosing the time derivative therefore we conclude we are free to choose the time derivative at the boundary of the time integral.

# B   Feynman rules with real and imaginary time mixing

First note that in the average-difference basis we no longer have the minus sign from the backwards contour, we have real propagators (with off-diagonal terms) and imaginary time propagators (with off-diagonal terms). So let's simplify and consider a fictitious theory with two sets of fields, one with real time (subscript 'M' for Minkowski) and one with imaginary time (subscript 'E' for Euclidean) and with off-diagonal propagators. In this theory,

$$
\langle f[q_M, q_E] \rangle = \int \mathcal{D}q_M \, \mathcal{D}q_E \, e^{iS_M - S_E} f[q_M, q_E]
$$

$$
= f\left[-i\frac{\delta}{\delta J_M}, -\frac{\delta}{\delta J_E}\right] \underbrace{\int \mathcal{D}q_M \, \mathcal{D}q_E \, e^{iS_{M,J} - S_{E,J}}}_{Z[V, J_M, J_E]}\Bigg|_{J_M = J_E = 0}, \tag{B.1}
$$

where

$$
S_{M,J} = \int dt \left\{\frac{1}{2}\dot{q}_M^2 - \frac{1}{2}m^2 q_M^2 - V(q_M) + J_M q_M\right\} = S_M + \int dt \, J_M q_M, \tag{B.2}
$$

$$
S_{E,J} = \int d\tau \left\{\frac{1}{2}\dot{q}_E^2 + \frac{1}{2}m^2 q_E^2 + V(q_E) + J_E q_E\right\} = S_E + \int d\tau \, J_E q_E. \tag{B.3}
$$

Our job is then to calculate $Z[V, J_M, J_E]$.

$$
Z[V, J_M, J_E] = e^{-i\int dt V\left(-i\frac{\delta}{\delta J_M}\right)} e^{-\int d\tau V\left(-\frac{\delta}{\delta J_E}\right)} \underbrace{\int \mathcal{D}q_M \, \mathcal{D}q_E \, e^{iS_{M,0,J} - S_{E,0,J}}}_{Z[J_M, J_E]}, \tag{B.4}
$$

where

$$
S_{M,0,J} = \int dt \left\{\frac{1}{2}\dot{q}_M^2 - \frac{1}{2}m^2 q_M^2 + J_M q_M\right\}, \tag{B.5}
$$

$$
S_{E,0,J} = \int d\tau \left\{\frac{1}{2}\dot{q}_E^2 + \frac{1}{2}m^2 q_E^2 + J_E q_E\right\}. \tag{B.6}
$$

From the discussion in the main body of the manuscript we know that $Z[J_M, J_E]$ will be of the form (up to normalisation):

$$
\begin{aligned}
Z[J_M, J_E] = \exp\bigg\{ &-\frac{i}{2}\int dt_1 \, dt_2 \, J_M(t_1) G_{MM}(t_1, t_2) J_M(t_2) \\
&+ \frac{1}{2}\int dt_1 \, d\tau_2 \, J_M(t_1) G_{ME}(t_1, \tau_2) J_E(\tau_2) \\
&+ \frac{1}{2}\int d\tau_1 \, dt_2 \, J_E(\tau_1) G_{EM}(\tau_1, t_2) J_M(t_2) \\
&+ \frac{i}{2}\int d\tau_1 \, d\tau_2 \, J_E(\tau_1) G_{EE}(\tau_1, \tau_2) J_E(\tau_2)\bigg\}. 
\end{aligned} \tag{B.7}
$$

Now let $J_M' = iJ_M$ and $J_E' = -J_E$ so that there are no confusing factors in the argument of the $V$'s in front. Coincidentally, this makes all factors in Eq. (B.7) the same and equal to $i/2$.

Then we can use the results from (B.11) from [57] to get

$$
\begin{aligned}
Z[V,J_M,J_E] = {} & \exp\left\{\frac{\mathrm{i}}{2}\int \mathrm{d}t_1\,\mathrm{d}t_2\, G_{MM}(t_1,t_2)\frac{\delta}{\delta q_M(t_1)}\frac{\delta}{\delta q_M(t_2)}\right\} \\
& \times \exp\left\{\frac{\mathrm{i}}{2}\int \mathrm{d}t_1\,\mathrm{d}\tau_2\, G_{ME}(t_1,\tau_2)\frac{\delta}{\delta q_M(t_1)}\frac{\delta}{\delta q_E(\tau_2)}\right\} \\
& \times \exp\left\{\frac{\mathrm{i}}{2}\int \mathrm{d}\tau_1\,\mathrm{d}t_2\, G_{EM}(\tau_1,t_2)\frac{\delta}{\delta q_E(\tau_1)}\frac{\delta}{\delta q_M(t_2)}\right\} \\
& \times \exp\left\{\frac{\mathrm{i}}{2}\int \mathrm{d}\tau_1\,\mathrm{d}\tau_2\, G_{EE}(\tau_1,\tau_2)\frac{\delta}{\delta q_E(\tau_1)}\frac{\delta}{\delta q_E(\tau_2)}\right\} \\
& \times \exp\left\{-\mathrm{i}\int \mathrm{d}t\, V(q_M)-\int \mathrm{d}\tau\, V(q_E)+\int \mathrm{d}t\, J'_M q_M+\int \mathrm{d}\tau\, J'_E q_E\right\}.
\end{aligned}
\tag{B.8}
$$

The factors that appear in the currents will cancel with the factors that are in the functional derivatives on $f$ above, in the end, for each power of $q$ we just need to add an external line. For the other Feynman rules we have (for a quartic potential):

$$
t_1 \,\text{———}\, t_2 \;= \mathrm{i}G_{MM}(t_1,t_2),
\tag{B.9a}
$$

$$
\tau_1 \,\text{———}\, \tau_2 \;= \mathrm{i}G_{EE}(\tau_1,\tau_2),
\tag{B.9b}
$$

$$
t_1 \,\text{———}\, \tau_2 \;= \frac{\mathrm{i}}{2}(G_{ME}(t_1,\tau_2)+G_{EM}(\tau_2,t_1)),
\tag{B.9c}
$$

$$
\tau_1 \,\text{———}\, t_2 \;= \frac{\mathrm{i}}{2}(G_{EM}(\tau_1,t_2)+G_{ME}(t_2,\tau_1)),
\tag{B.9d}
$$

$$
\times_t \;= -\mathrm{i}\lambda\int \mathrm{d}t\,,
\tag{B.9e}
$$

$$
\times_\tau \;= -\lambda\int \mathrm{d}\tau\,.
\tag{B.9f}
$$

## C  Additional 1-loop checks

If the picture described in the main body of the text is to hold then the same counterterms as defined in (4.18) should cancel all divergences regardless of whether we include them in the free or the interaction Hamiltonian. Further, the resummation prescribed in (4.23) should still work, which implies a very particular structure of the 1-loop corrections. A full proof to all orders in perturbation theory is still lacking but in this appendix we test it for the remaining propagators.

### C.1  Corrections to $G_{\text{dif,ave}}$

The only diagram that contributes to this is:

$$\text{(diagram)} \qquad \text{(C.1)}$$

Via a straightforward evaluation of this diagram and expanding the tree-level propagator in a similar manner to (4.6) we get:

$$G_{\text{dif,ave}}^{\text{1-loop}}(\boldsymbol{p}, t_1, t_2) = -\frac{\lambda I_\beta + 4\delta m^2}{8E_{\boldsymbol{p}}^3} \Theta(t_2 - t_1)\Big(E_{\boldsymbol{p}}(t_2 - t_1)\cos\big[E_{\boldsymbol{p}}(t_1 - t_2)\big] + \sin\big[E_{\boldsymbol{p}}(t_1 - t_2)\big]\Big), \quad \text{(C.2)}$$

which has the required structure.

## C.2   Corrections to $G_{\text{ave,E}}$

In this case we get two diagrams:

$$\text{(diagrams)} \qquad \text{(C.3)}$$

It is not immediately obvious that what we get via direct computation of the diagrams and the expansion of the propagator match. However, after some tedious trigonometric simplications one finds

$$
\begin{aligned}
G_{\text{ave,E}}^{\text{1-loop}}(\boldsymbol{p}, t_1, \tau_2) = {} & \frac{\lambda I_\beta + 4\delta m^2}{32 E_{\boldsymbol{p}}^3}\operatorname{csch}\left(\frac{E_{\boldsymbol{p}}\beta}{2}\right)\left(2\mathrm{i}\cos\left[E_{\boldsymbol{p}}\left(t_1 - t_0 + \mathrm{i}\tau_2 - \frac{\mathrm{i}\beta}{2}\right)\right]\right. \\
& + \mathrm{i}E_{\boldsymbol{p}}\beta\cos\left[E_{\boldsymbol{p}}\left(t_1 - t_0 + \mathrm{i}\tau_2 - \frac{\mathrm{i}\beta}{2}\right)\right]\coth\left(\frac{E_{\boldsymbol{p}}\beta}{2}\right) \\
& \left. + 2\left(\mathrm{i}t_1 - \mathrm{i}t_0 + \frac{\beta}{2} - \tau_2\right)\sin\left[E_{\boldsymbol{p}}\left(t_1 - t_0 + \mathrm{i}\tau_2 - \frac{\mathrm{i}\beta}{2}\right)\right]\right).
\end{aligned} \qquad \text{(C.4)}
$$

Which has the expected structure. It is worth noting that this diagram seems to have genuine secular behaviour, however, it does not by itself correspond to a physical observable therefore this is of no major concern. Calculating this and the next diagram is useful merely as a way to organise the perturbative expansion.

## C.3   Corrections to $G_{E,E}$

There is only one diagram to consider:

$$\text{(diagram)} \qquad \text{(C.5)}$$

Once more it is not entirely trivial to manipulate the trigonometric expressions, nonetheless, the final answer is:

$$
\begin{aligned}
G_{E,E}^{\text{1-loop}}(\boldsymbol{p}, \tau_1, \tau_2) =& \mathrm{i}\frac{\lambda I_\beta + 4\delta m^2}{32 E_{\boldsymbol{p}}^3} \operatorname{csch}\left(\frac{E_{\boldsymbol{p}}\beta}{2}\right)\left(E_{\boldsymbol{p}}(\beta - 2|\tau_1 - \tau_2|) \sinh\left[E_{\boldsymbol{p}}\left(|\tau_1 - \tau_2| - \frac{\beta}{2}\right)\right]\right.\\
&+ \cosh\left[E_{\boldsymbol{p}}\left(|\tau_1 - \tau_2| - \frac{\beta}{2}\right)\right]\left.\left(2 + \beta E_{\boldsymbol{p}} \coth\left(\frac{E_{\boldsymbol{p}}\beta}{2}\right)\right)\right).
\end{aligned}
\tag{C.6}
$$

All the above comments apply: the divergence structure is what we desire and despite not being physical it is still useful in perturbation theory.

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
