# Peer review of "The Propagator Matrix Reloaded"

_SciPost Physics, doi:SciPost Phys. Core 6, 019 (2023)_

## Round 1 · Referee Report · Anonymous (Referee 4) · 2022-11-8

Strengths

1 - this manuscript discusses a delicate point in quantum field theory at finite temperature (in equilibrium),;namely that, to be in equilibrium, interactions must be present in the operator density that defines the thermal ensemble. This implies that it is not allowed to ignore interactions in the remote past

2 - it provides a novel practical way to take into account correctly these interactions from the density operator

Weaknesses

1 - this issue has been discussed in several anterior works

2 - another practical way of including the contribution from the interactions in the density operator has been known for a long time

3 - it seems no well known existing paper is incorrect because of having ignored this issue, so this problem is more academic than a real practical concern

Report

This manuscript deals with the perturbative expansion in relativistic quantum field theory in equilibrium at finite temperature, and more specifically with how to deal with the interactions in the initial state (i.e., with the interactions coming from the canonical density operator exp(-\beta H)).

Let me recall here what was known prior to this work:

1. When formulating the theory in spacetime, the time integrations should be performed on a "thermal contour", made of two branches on the real axis, and one branch in the imaginary direction. The branch in the imaginary direction encodes the interactions in the initial density operator. Neglecting its contribution would amount to using $exp(-\beta H_{free})$ as the density operator, which would violate the KMS identities, as it would amount to suddenly turning on the interactions at the initial time.

2. Several approaches have been used in the past to deal with this issue:

2.a. Take the initial time to $-\infty$ and somehow argue that the contribution of the branch in the imaginary direction vanishes in this limit. This is an **incorrect** argument, since one may prove easily that nothing depends on the initial time for a system in equilibrium (in a system in equilibrium, no measurement can tell the time at which it was prepared in equilibrium).

2.b. Modify the density operator so that it goes to the free one when the initial time goes to $-\infty$ and simultaneously modify the propagators so that KMS relation is nevertheless always true. In this limit, the branch in the imaginary direction does not contribute.

2.c. Keep the initial time finite and keep the full density operator. One may show that:

- no observable depends on the initial time

- observables (including the contribution of the piece of the time contour in the imaginary direction) can be obtained by using a $2\times2$ matrix propagator, provided one uses the substitution $f(E_p) -> f(|p_0|)$. These two prescriptions for the argument of the particle distribution are equivalent when one considers a single free propagator in isolation, but may lead to a difference when propagators are multiplied.

3. I do not know of calculations that use the $3\times 3$ propagator advocated in the present manuscript. I also do not know of any important existing paper that has been proven incorrect because its authors have ignored the issues discussed in the present manuscript.

My impression is that the present manuscript is correct, but not very useful in practice because it merely (re)explains issues that have been known for quite some time. Moreover, the $3\times3$ matrix formalism advocated here does not seem to bring different results than the old "recipes" that have been used for almost 40 years (and are somewhat simpler to implement, although of course this is a very subjective opinion).

In my opinion, this manuscript could nevertheless be published because it sheds a new light on an old issue, even if it will probably not change significantly the way people do calculations at finite temperature.

  • validity: good
  • significance: low
  • originality: good
  • clarity: good
  • formatting: good
  • grammar: good

Author:  João Melo  on 2022-11-11  [id 3009]

(in reply to Report 1 on 2022-11-08)
Category:
answer to question
reply to objection

First of all I would like to thank you for your very insightful comments.

I was indeed aware of the progress you mention and I even referenced it in my work but, perhaps, I was not explicit enough in my discussion so allow me to clarify some of the points raised.

In reference [51] of my manuscript we have the most careful treatment of the $f(E_p)\to f(p_0)$ prescription I found. In it, the authors show that this originally rather ad-hoc prescription (similar to the 2b method you describe) was actually equivalent to resumming the mass loops into the original propagator as I described at the end of section 4. This is completely in line with the findings in my paper, the mass resummation found via a less careful 2x2 method is the same as the more careful 3x3 method. So, in practice, the 2x2 method is able to capture all of the physics of a 2-point function, as long as this resummation holds to all others in perturbation theory. It is less mathematically rigorous but it works in practice.

In a follow-up [52] the authors argue that indeed this is all of the physics of the cross-terms between real and imaginary time even for higher point functions (what they call the vertical part of the contour). However, the calculations performed in section 5 of my paper show this argument cannot hold. In it I calculate a simple 4-point function using the 3x3 method and I found a contribution from these cross-terms which cannot be absorbed into a mass resummation. Indeed the only remaining contribution is that of the cross terms.

In summary, it is true that $f(E_p)\to f(p_0)$can capture a lot of the physics of the interactions in the initial state but it cannot capture all of them. In particular it gives the wrong answer for higher point functions.

Regarding possible errors in past papers due to these findings, any paper which restricts their attention to 2-point functions should still hold. However, calculations of higher point functions which do not take into account these effects may be incorrect.

I would of course be perfectly happy with making this discussion more explicit in the paper to avoid further confusion.

I hope I managed to address your concerns, do not hesitate in contacting me if I missed anything.

---

## Round 1 · Referee Report · Anonymous (Referee 5) · 2022-11-22

Report

The Author has considerably improved the presentation of his work, in particular on the existing state of the art. A new calculation section on four-point functions has been added, adding to the clarity of the overall message, and adding to the originality of the paper.

As it stands, this work does not represent a significant advance in the understanding of real-time path-integral formalisms. Although practitioners of the Keldysh formalism may not learn much from it, this work (and its significant level of detail in the calculations) can nevertheless be useful because to beginners or people seeking to perform similar calculations.

Therefore, I can recommend publishing this work in SciPost Physics Core.

---

## Round 1 · Author Response

Taking into account the editor and the referee's comments the manuscript has changed significantly. I now explicitly address the most common ways this issue has been addressed and avoided and discuss their issues, including the differences with the non-relativistic case that one referee pointed out.

The central message is nevertheless unchanged: it is essential to consider the effect of interactions in the far past and this issue has been widely misunderstood and under-examined in the literature.

These changes are most evident in the introduction, conclusion and in the discussion following equation (4.23).

Further, I have added a section calculating the tree-level equal time 4-point function. This calculation makes it clear that an approach which only uses the 2x2 matrix cannot be correct as it will certainly fail to reproduce these effects.

---

## Round 1 · List of Changes

- Introduction, conclusion and abstract changed substantially. Now they exhibit a new overarching narrative which a much more explicit comparison with the previous literature

- References added to fully cement the discussion

- Added section 5 which calculates the tree-level equal time 4-point function to give further evidence for the conclusion

- Added discussion after Eq. (4.23) comparing my approach with the most common approaches in the previous literature

- Minor typos corrected

---

## Editorial Decision

published